# Effect of a Sustainable Air Heat Pump System on Energy Efficiency, Housing Environment, and Productivity Traits in a Pig Farm

**Myeong Gil Jeong** [1,†], **Dhanushka Rathnayake** [1,†], **Hong Seok Mun** [1],
**Muhammad Ammar Dilawar** [1], **Kwang Woo Park** [2], **Sang Ro Lee** [2] **and Chul Ju Yang** [1,*]

[1] Animal Nutrition and Feed Science Laboratory, Department of Animal Science and Technology, Sunchon National University, Suncheon 57922, Korea; wjdaudrlf13@naver.com (M.G.J.); dhanus871@gmail.com (D.R.); mhs88828@nate.com (H.S.M.); ammar_dilawar@yahoo.com (M.A.D.)
[2] WP Co., Ltd., Suncheon 58023, Korea; pkw9872@naver.com (K.W.P.); skylife37@naver.com (S.R.L.)
[*] Correspondence: yangcj@scnu.kr; Tel.: +82-61-750-3235
[†] This author contributed equally to this work as co-first author.

**Abstract:** High electricity consumption, carbon dioxide ($CO_2$), and elevated noxious gas emission in the global livestock sector have a negative influence on environmental sustainability. This study examined the effects of a heating system using an air heat pump (AHP) on the energy saving, housing environment, and productivity traits of pigs. During the experimental period of 16 weeks, the internal temperature was found to be higher ($p < 0.05$) in the AHP house than in the conventional house. Moreover, the average electricity consumption and $CO_2$ emission decreased by approximately 40 kWh and 19.32 kg, respectively, in the AHP house compared to the house with the conventional heating system. The average $NH_3$ and $H_2S$ emissions were significantly lower in the AHP house ($p < 0.05$) during the growth stages. The AHP and conventional heating systems did not have a significant influence ($p > 0.05$) on the average ultra-fine dust ($PM_{2.5}$) and formaldehyde level fluctuations. Furthermore, both heating systems did not show a significant difference in the average growth performance of pigs ($p > 0.05$), but the weight gain tended to increase in the AHP house. In conclusion, the AHP system has great potential to reduce energy consumption, greenhouse gas (GHG) emissions, and noxious gas emissions by providing economic benefits and an eco-friendly renewable energy source.

**Keywords:** air heat pump; carbon dioxide; formaldehyde; electricity consumption; ultra-fine dust

## 1. Introduction

Various energy problems have been identified in the global agriculture sector not only for economic reasons, but also for sustainable ecological persistence [1–3]. This is due to the diminishing fossil fuel reserves and increasing energy prices worldwide [4,5]. In addition, excessive greenhouse gas emissions affect biodiversity degradation through global warming [6]. Furthermore, the increase of global $CO_2$ emissions into the atmosphere is expected to lead to a temperature increase from 1.1 to 6.4 °C by the end of the 21st century [7]. Fossil-fuel burning is the major contributor of $CO_2$ emissions, and the atmospheric $CO_2$ concentration has been enhanced by 31% since 1750, with an average annual increase by 1.5 ppm over the past decades [8]. Beside deforestation and excessive arable land utilization, fossil-fuel combustion is responsible for 90% of $CO_2$ emissions into the environment [9].

In the global livestock sector, pigs have an inefficient thermoregulation process for dissipating heat from their bodies. Their maximum voluntary feed intake (VFI) ranges from 19 to 25 °C and tends

to decrease above 25 °C [10]. $NH_3$ and methane byproducts released by pigs, together with dust, affect the air quality and are considered important parameters in pig houses [11]. The emission of noxious gas from the livestock sector is one of the major problems; it exerts negative impacts on the environment and accounts for approximately 75–80% of $NH_3$ emissions in developed nations in the world [12]. Moreover, a combination of both $NH_3$ and $H_2S$ adversely affects the pig industry [13,14] owing to the direct harmful impact on both animals' and workers' welfare [15]. Dust can penetrate the respiratory organs easily owing to its smaller particle size. Super-fine dust particles less than <1 μm are the most harmful and cause pulmonary diseases [16]. Therefore, essential steps are needed to improve the housing environment by reducing noxious gas emissions, dust concentration, and environmental pollution. Moreover, due to the presence of an abundant renewable resource capacity, South Korea has the potential of utilizing them efficiently to mitigate the problems arising through high energy consumption, thus finding effective solutions for those challenges and the energy distribution process across the various geographical areas [17].

The air heat source pump system has the potential to conserve high-grade energy and allow the effective use of low-grade energy, as well as to provide energy savings and storage [18]. Other than the energy savings, it can reduce $CO_2$ emissions and is consistent with efficient structural compaction [19].The theoretical and experimental performance and effectiveness of the air heat pump were investigated by previous studies [20–23]. However, there are no publications available on the effects of utilization of air heat pump systems on energy efficiency, housing environments, and productivity traits in livestock sectors. Owing to the environmentally friendly and sustainable source, an air heat pump system can be introduced as an alternative energy system for conventional methods. Therefore, this study compared a conventional electric heating system and an air heat pump (AHP) system for the energy savings, housing environment ($NH_3$, $H_2S$, fine dust, formaldehyde), and productivity traits of pigs.

## 2. Materials and Methods

### 2.1. Experimental Period and House

The performance of the air pump heating system in a pig house was evaluated for 16 weeks (weaning period, four weeks; growing period, six weeks; finishing period, six weeks) in winter from 2 December 2019 to 2 April 2020 at the Sunchon National University Experimental Farm, South Korea. The pig house consisted of two separate rooms (3 m × 8 m) that were subdivided into 10 pens for individual replication. Two east-facing rooms were contained in the pig house. The room on the south side was considered the control house, which was connected to a conventional electric heating system. The air pump heating system was connected to the other north-facing room (Figure 1). An outdoor unit draws heat in from outside, and thereafter, blows it over a heat exchanger coil. The heat thus generated from the compressor is then transferred through an internal plastic tube with small pores that enable the uniform distribution of the heating pattern inside the house. Finally, the cold liquid vapor coolant mixture enters back into the outdoor unit to be heated once again. The conventional pig house was connected with heating lamps; the heights of these were maintained according to the growth phase of the pigs. The outside walls of the pig house were made from brick plastered on both sides. The floor was installed with a plastic slat, and the slurry was removed daily. The environmentally controlled pig houses' inside temperature ventilation processes were controlled automatically. Moreover, we maintained similar internal temperature settings according to each growing phase, covering both the conventional electric heating house and the AHP installed house to compare the inside temperature fluctuations, energy efficiency, noxious gas emission, ultra-fine dust, and formaldehyde concentration between the two experimental houses. Throughout the experiment, all animals received a commercial basal diet and had access to water ad libitum.

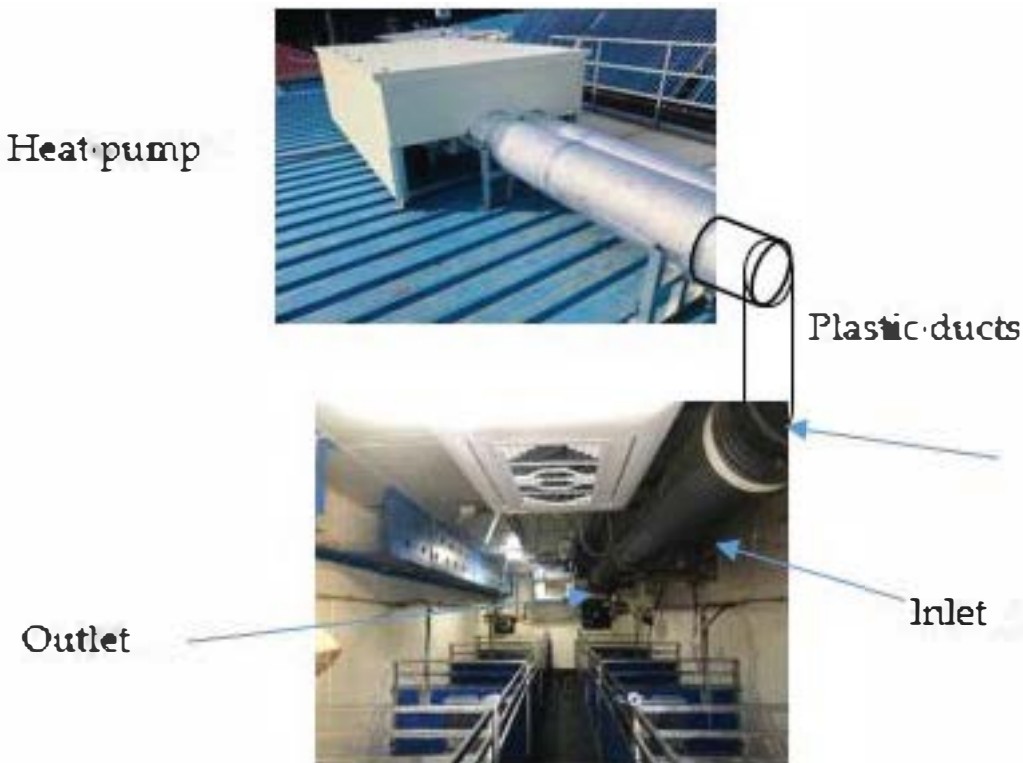

**Figure 1.** Schematic diagram of the air heat pump (AHP) system for the pig house.

## 2.2. Description of the Air Heat Pump System

The air heat pump (AHP) (model: BW1450M9S, LG Electronics Inc., Seoul, South Korea) was installed and connected to a pig house according to a slight modification of the procedures recommended by previous studies [24,25]. The major components of the air heat pump system were an air inlet, inhale chamber, air heat pump compressor, discharge chamber, and air-circulating pipes (Figure 2).

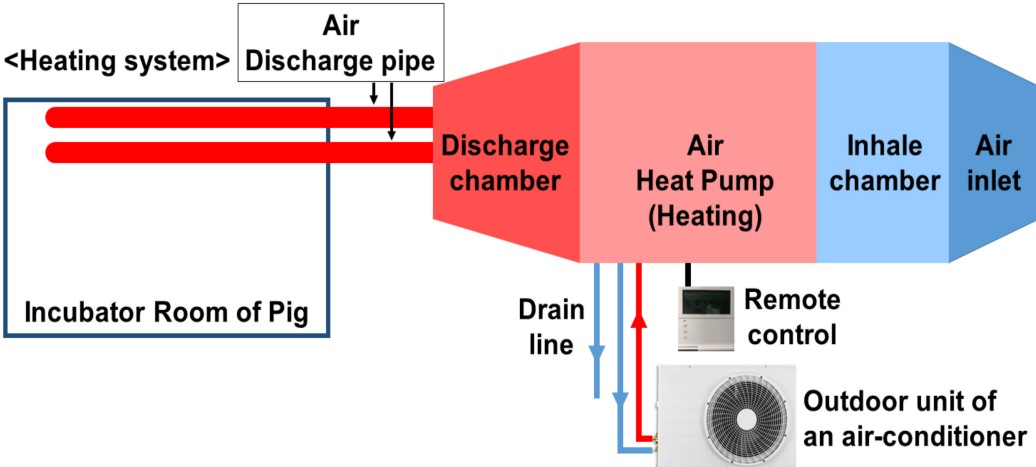

**Figure 2.** Outline of the air heat pump system.

The power was supplied through a three-phase four-wire system (380 V, 60 Hz). The estimated minimum and maximum heating ability values were 5.2 and 20 kW, respectively. The evaporator coil system of the heat pump (HP) system could dehumidify and cool the extracted hot and wet air. The absorbed fresh and purified air was heated by a condenser, and the circulating fluid was the refrigerant R410A. The inlet fan was controlled thermostatically, and the temperature level was

maintained according to its speed. The extraction fan speed was controlled manually. The required power for the operation of the compressor was 1.1 kW; the required level increased to 4.3 kW when the fans were operating. The coefficient of performance was 4.3 for the heating process when the reference temperature values were used: external air at 6.0 °C, evaporation at −4.0 °C, and condensation at 45 °C.

### 2.3. Measurement and Analysis

The temperatures of the control and air heat pump houses were determined using eight-bit Smart Sensors (model: SMT-75, Seoul, South Korea). Temperature data were taken from the ceiling at the entry (close to door), center and back of the pig houses at 10 cm above the slatted floor (lower point), and 10 cm below the ceiling level. All measuring equipment was connected to a data logger system (CR10X data logger, Campbell Scientific Inc., Edmonton, AB, Canada) to record the data for every hour. The recording equipment was properly designed for an auto-restart process to prevent data losses due to power failures. A digital hygrometer (Electronic Digital Hygrometer HTC-1, Jinggoal International Ltd., Guangdong, China) was used to evaluate the humidity level inside the both pig houses.

The coefficient of performance (COP) of the heat pump was evaluated using the following formula [26]:

$$COP = \frac{\sum \dot{Q}}{\sum \dot{W}} \tag{1}$$

where $\dot{Q}$ is the useful heat extracted from the heat pump (condenser) (kW) of the air heat pump; $\dot{W}$ is the power consumption (kW).

The daily electricity consumption of both the conventional and air heat pump house was measured based on the electricity consumption units recorded by individually installed meters (Model: LD 1210Ra-040, LSis, Seoul, South Korea). The daily electricity cost of each house was calculated according to the current electricity cost in South Korea (Korea Electric Corporation, KEPCO, September 2020 (1 kWh electricity = 39.2 South Korean won, and 39.2 South Korean won = 0.033 USD)). In addition, $CO_2$ emissions were determined in $kgCO_2e$ (1 kWh = 0.483 kg $CO_2$ equivalent) [27] according to the electricity consumption in both pig houses.

$NH_3$ and $H_2S$ gas concentrations were evaluated every day at 8:00 am at the entry, center, and back positions at approximately 30 cm above the slatted floor using a Gastec (model GV-100) gas sampling pump (Gastec Corp., Kanagawa, Japan) and gas detector tubes: No. 3L (0.5–78 ppm, Gastec Corp., Kanagawa, Japan) for $NH_3$ and 4LT (0.05–4 ppm, Gastec Corp., Kanagawa, Japan) for $H_2S$. The $NH_3$ gas emission was expressed in ppm, and the $H_2S$ level was expressed in ppb in both pig houses. The ultra-fine dust concentration and formaldehyde level were measured every day during the experimental period at 8:00 am at the entry, center, and back of each pig house using a Smart Sensor air quality model (model:AR830A-2, Huipu Opto-Electronic Instrument (Zhenjiang) Co., Ltd., Jian, China) at 10 cm above the floor.

The body weight gain, feed intake, and feed conversion ratio (FCR) were measured during the weaning, growing, and finishing periods. The body weight gain was evaluated by dividing the weight difference of the starting and finishing weight by each experimental period. The feed intake was measured every week by weighing the feed weight immediately before the body weight measurement. The FCR was calculated by dividing the feed intake by the average daily gain.

### 2.4. Statistical Analysis

The inside room temperature, noxious gas emission, ultra-fine dust concentration, and formaldehyde level in the experimental houses were evaluated using PROC GLM of the statistical analysis system (version 9.1, SAS Institute Inc., Cary, NC, USA). The data are reported as the mean ± standard error of the means (SEM). A *p*-value < 0.05 was considered significant.

## 3. Results

### 3.1. Room Temperature and Coefficient of Performance (COP)

As shown in Table 1, the temperature level was increased significantly ($p < 0.05$) in the house with the air heat pump compared to the control house. The highest and lowest temperatures in the AHP house were 26.1 and 19.9 °C, respectively (Figure 3). The COP value was reduced when the external temperature was decreased during the weaning period.

**Table 1.** Effect of the air heat pump system on inside temperature and coefficient of performance (COP).

| Periods | External Temp. (°C) | Control (°C) | AHP (°C) | SEM | *p*-Value | Average COP |
|---|---|---|---|---|---|---|
| Weaning | 4.5 | 24.7 [b] | 26.1 [a] | 1.84 | <0.0001 | 3.86 |
| Growing | 6.1 | 20.4 [b] | 22.8 [a] | 1.88 | <0.0001 | 3.98 |
| Finishing | 9.7 | 19.9 [b] | 21.1 [a] | 1.14 | <0.0001 | 4.12 |
| Average | 7.1 | 21.3 [b] | 23.0 [a] | 2.59 | <0.0001 | 4.07 |

[a, b] means that values with different superscripts within same row are significantly different ($p < 0.05$).

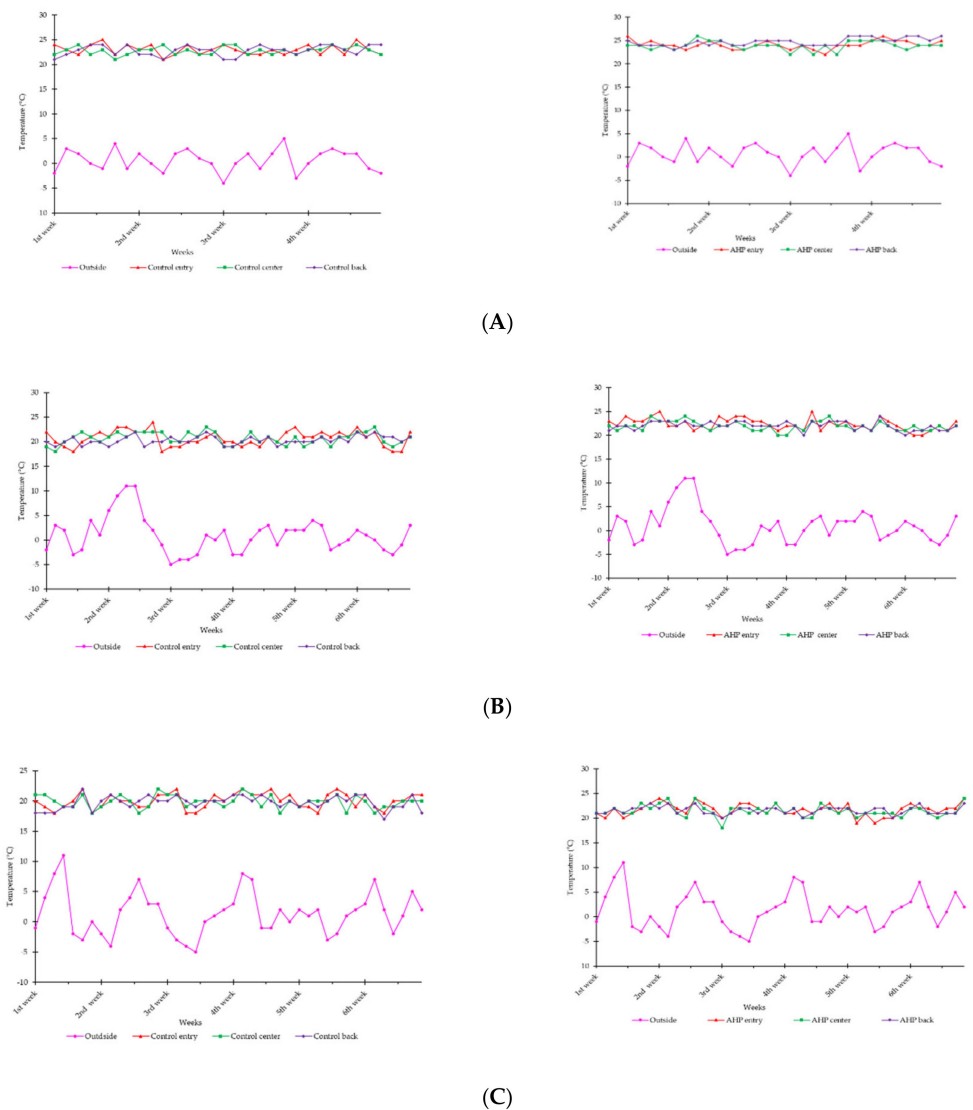

**Figure 3.** Temperature profile for control (conventional electric heating system) versus air heat pump (AHP) housing system during winter, at the entry, center, and back position of the rooms (average mean value at lower and upper positions). (**A**) Temperature profile during weaning period (four weeks). (**B**) Temperature profile during growing period (six weeks). (**C**) Temperature profile during finishing period (six weeks).

### 3.2. Electricity Consumption, CO₂ Emissions, and Cost Savings

Table 2 lists the daily electricity consumption and cost analysis per day in both experimental houses. The daily electricity consumption during the weaning, growing, and finishing periods were decreased in the AHP house. The decrease in average daily electricity consumption in the AHP house was 63.5%.

**Table 2.** Energy consumption and heating costs of the pig houses during the various growth periods.

| Period | Electricity Use (kWh/d) | | | $CO_2$ Emission (kg) | | | Cost Savings (USD) |
|---|---|---|---|---|---|---|---|
| | Control | AHP | Reduced | Control | AHP | Reduced | |
| Weaning | 108 | 33 | 75 | 52.16 | 15.94 | 36.22 | 97.02 |
| Growing | 60 | 30 | 30 | 28.98 | 14.50 | 14.48 | 38.80 |
| Finishing | 35 | 9 | 26 | 16.91 | 4.35 | 12.56 | 33.63 |
| Average | 63 | 23 | 40 | 30.43 | 11.11 | 19.32 | 51.74 |

Electricity consumption was determined using an electric meter installed for each house for every day (8:00–20:00) and night (20:00–8:00), and was summed per day. Carbon dioxide ($CO_2$) emissions were evaluated based on electricity consumption per day. The used conversion factor was 1 kWh = 0.483 kg $CO_2$ emissions [27]. Cost was estimated according to electricity consumption per day. The current value of 1 kWh electricity = 39.2 South Korean Won, and 39.2 South Korean Won = 0.033 USD was used (KEPCO, September 2020).

Consequently, a significant decline of the daily electricity cost was also observed in the AHP house relative to the control house. During the finishing period, the electricity consumption was reduced drastically in the AHP house, and the reduced average daily electricity cost was 63.6% compared to the control pig house. The total $CO_2$ emission was decreased in the AHP-installed house, and the average daily reduction was 63.49% compared to the control house.

### 3.3. $NH_3$, $H_2S$, Ultra-Fine Dust ($PM_{2.5}$), and Formaldehyde Level

As listed in Table 3, $NH_3$ and $H_2S$ emissions during the weaning, growing, and finishing periods were significantly lower ($p < 0.05$) in the AHP house compared to the control house. The average $NH_3$ emissions were reduced by 61%, and the average $H_2S$ level was decreased by 45% in the AHP house.

**Table 3.** Effect of the air heat pump system on the $NH_3$ and $H_2S$ emissions in the pig house.

| Item | Periods | Control | AHP | SEM | *p*-Value |
|---|---|---|---|---|---|
| $NH_3$ (ppm) | Weaning | 0.05 [a] | 0.01 [b] | 0.02 | <0.0001 |
| | Growing | 0.63 [a] | 0.16 [b] | 0.33 | <0.0001 |
| | Finishing | 2.27 [a] | 0.97 [b] | 0.45 | <0.0001 |
| | Average | 1.10 [a] | 0.42 [b] | 0.78 | <0.0001 |
| $H_2S$ (ppb) | Weaning | 0.00 | 0.00 | 0.00 | - |
| | Growing | 0.00 | 0.00 | 0.00 | - |
| | Finishing | 5.14 [a] | 2.81 [b] | 0.56 | <0.0001 |
| | Average | 1.93 [a] | 1.06 [b] | 0.94 | <0.0001 |

[a, b] means that values with different superscripts within the same row are significantly different ($p < 0.05$).

Table 4 lists the ultra-fine dust concentration ($PM_{2.5}$) and formaldehyde concentration due to the air heat pump system. There were no significant differences in the $PM_{2.5}$ dust concentration between the two houses. On the other hand, the dust concentration tended to decrease during all periods, and the average reduction was 6.5% in the AHP house compared to the control house. During the growing and finishing periods, the formaldehyde concentration was significantly lower ($p < 0.05$) in the AHP house.

**Table 4.** Effect of the air heat pump system on ultra-fine dust ($PM_{2.5}$) and formaldehyde concentration in the pig house.

| Item | Periods | Control | AHP | SEM | *p*-Value |
|---|---|---|---|---|---|
| Ultra-fine dust ($PM_{2.5}$) ($\mu g/m^3$) | Weaning | 28.39 | 24.74 | 16.60 | 0.46 |
| | Growing | 29.03 | 26.33 | 23.71 | 0.64 |
| | Finishing | 21.14 | 21.69 | 14.22 | 0.87 |
| | Average | 25.84 | 24.14 | 18.80 | 0.54 |
| Formaldehyde (ppm) | Weaning | 0.06 | 0.05 | 0.02 | 0.26 |
| | Growing | 0.08 [a] | 0.05 [b] | 0.04 | 0.03 |
| | Finishing | 0.22 [a] | 0.13 [b] | 0.19 | 0.02 |
| | Average | 0.12 | 0.10 | 0.12 | 0.27 |

[a, b] means that values with different superscripts within the same row are significantly different ($p < 0.05$).

### 3.4. Effect of the Air Pump Heating System on the Productivity Traits of Pigs

Table 5 lists the results of the growth performances of pigs during their weaning, growing, finishing, and average values. The body weight gain, feed intake, and feed conversion ratio (FCR) did not significantly differ ($p > 0.05$) among the control and AHP-installed house.

**Table 5.** Effect of the air pump heating system on the productivity parameters of pigs.

| Item | Control | AHP | SEM | *p*-Value |
|---|---|---|---|---|
| **Weaning period (0–4 weeks)** | | | | |
| Initial weight (kg) | 8.56 | 8.29 | 3.17 | 0.86 |
| Final weight (kg) | 25.81 | 25.87 | 5.66 | 0.98 |
| Weight gain (kg) | 17.26 | 17.58 | 3.01 | 0.82 |
| Feed intake (kg) | 33.48 | 33.79 | 7.45 | 0.93 |
| FCR (Feed/gain) | 1.95 | 2.01 | 0.60 | 0.83 |
| **Growing period (4–10 weeks)** | | | | |
| Initial weight (kg) | 25.81 | 25.87 | 5.66 | 0.98 |
| Final weight (kg) | 70.77 | 66.10 | 7.80 | 0.22 |
| Weight gain (kg) | 40.23 | 43.96 | 4.27 | 0.03 |
| Feed intake (kg) | 100.93 | 96.66 | 4.63 | 0.54 |
| FCR (Feed/gain) | 2.25 | 2.44 | 0.41 | 0.36 |
| **Finishing period (10–16 weeks)** | | | | |
| Initial weight (kg) | 70.77 | 66.10 | 7.80 | 0.22 |
| Final weight (kg) | 113.43 | 107.97 | 7.18 | 0.13 |
| Weight gain (kg) | 41.87 | 42.66 | 4.64 | 0.73 |
| Feed intake (kg) | 150.81 | 151.48 | 4.26 | 0.92 |
| FCR (Feed/gain) | 3.53 | 3.69 | 0.49 | 0.51 |
| **Average (0–16 weeks)** | | | | |
| Initial weight (kg) | 8.56 | 8.29 | 2.80 | 0.96 |
| Final weight (kg) | 113.43 | 107.97 | 7.18 | 0.13 |
| Weight gain (kg) | 99.68 | 104.88 | 5.46 | 0.06 |
| Feed intake (kg) | 285.23 | 281.93 | 3.23 | 0.82 |
| FCR (Feed/gain) | 2.71 | 2.83 | 0.23 | 0.30 |

### 3.5. Estimation of the Installation and Annual Operational Costs

As shown in Table 6, the initial investment for the air heat pump system was comparatively higher than for the conventional electric heating system. Nevertheless, the AHP system gained a lower annual operational cost, higher life span, and shorter payback period.

**Table 6.** Installation and operational costs of the air heat pump and conventional electric heating system.

| Item | Control | AHP | |
|---|---|---|---|
| Installation cost (USD) | 1288 | 5000 | |
| Life span | 5 years | 15 years | As per company instruction |
| Annual operational cost (USD) | 4323 | 394 | |
| Savings (USD) | - | 3929 | |
| Payback period (Y) | >useful life | 4.1 | |
| Depreciation time | 5 years | 15 years | |

Annual operational cost was evaluated according to annual electricity consumption (131,026 kWh) and price (0.033 USD/kWh).

## 4. Discussion

Proper temperature maintenance inside a pig house is essential to prevent pigs from cold shock and ensure their optimal growth. In this study, the inside temperature of the AHP house was greater than in the conventional electric heating system. We speculated that the AHP system could distribute a uniform heat pattern more continuously inside the house than the conventional electric heating system due to the high COP value and lower running time period. The calculated average COP of this study was 4.07, which is lower than the values observed by Riva et al. [28] during the heating phase, but it was higher than the experiment conducted by Ji et al. [29] using an air heat pump for domestic heating purposes. In contrast, Zang et al. [30] reported that the COP value tends to increase with decreasing external temperature because the evaporator of the heat pump interacts continuously with hot air circulation during the heating phase. Nevertheless, some studies have reported that the efficiency of the AHP system tends to decrease when exposed to extreme temperature levels [31–33].

Renewable energy sources are abundant, have low cost, and are environmentally safe. In the present study, the AHP system showed lower electricity consumption, $CO_2$ emissions, and electricity cost relative to the conventional electric heating system. To the authors' knowledge, only one study has evaluated the effects of an air pump system on energy savings and housing environment in pig breeding house [28]. Riva et al. [28] reported that an AHP house could save 11% of the total energy consumption compared to a control house connected to an LPG boiler house. Similarly to the present results, Wu [34] concluded that the AHP system is a more efficient environmental safety system than conventional heating techniques, and can be introduced to minimize the depletion of energy resources. The low electricity consumption in the AHP system might be due to the high COP value, which has the potential to distribute unvarying heat inside the experimental house.

Rabczak et al. [35] reported that the emission of $CO_2$ to the atmosphere could be reduced by 40% with an air heat pump system compared to a local gas furnace system or particular heating system that is provided for specific geographical areas. Furthermore, decreased $CO_2$ emissions and energy savings have been investigated in response to the air pump heating system in buildings [36–39]. In addition to the increasing feed cost, energy prices have a huge impact on the productivity of the global pig industry, including in South Korea. In the present study, the electricity cost decreased during each growth period in the AHP-installed house compared to the conventional system.

According to the International Commission of Agricultural and Bio-Systems Engineering, CIGR (2002) [40], the recommended maximum $NH_3$ concentration is 20 ppm. In the livestock sector, pig growth was slowed by 12% to 30% in intensive swine buildings because of the elevated $NH_3$ concentration [11]. An improper ventilation system and high concentrations of $NH_3$, $H_2S$, and $CO_2$ lead to poor air quality inside pig houses. In the present study, the concentrations of both $NH_3$ and $H_2S$ were significantly lower in the AHP house, which is in agreement with the results of a previous study [28]. The lower noxious gas concentration may have occurred due to the increased fresh outdoor air temperature due to the compressor, as well as the subsequent dilution of $NH_3$ and $H_2S$ levels in the pig house.

Takai et al. [41] reported that the dust concentration in swine houses tends to increase in winter compared to summer. Automotive exhaust and various urea–formaldehyde products are the

sources of formaldehyde formation inside houses. Exposure to 0.3 to 50 ppm will depreciate lung compliance [42]. In the present study, during the growing and finishing periods, air contamination with formaldehyde was lower, possibly due to proper air circulation inside the AHP house. On the other hand, the relationship between the dust concentration, formaldehyde level, and installation of an AHP system is unclear. Further research on dust and formaldehyde fluctuations from the utilization of renewable energy sources will be needed.

According to Riva et al. [28], the production parameters, including feed intake, weight gain, and feed conversion ratio, increased significantly in a pig house operating with the AHP system compared to one operating with an LPG gas system. The accumulation of high concentrations of fumes in an LPG house may reduce their voluntary feed intake because of the poor housing environmental conditions. Nevertheless, in our study, there were no significant differences in the productivity parameters, but the weight gain tended to increase in the AHP house during the growth stages. Therefore, further study on the productivity parameters when using the AHP system in the livestock sector will be needed.

Owing to the high COP value of the AHP system, the annual operational cost was reduced by 91% compared to the control heating system. Wu [34] reported that the air heat pump reduced electricity consumption by 46 $kWh/m^2$. Consequently, it reduces the electricity costs. Islam et al. [43] reported that the installation cost for a renewable geothermal heat pump is considerably more expensive than for a renewable AHP system, and both systems had lower annual operating costs than the electric heating system. The payback period tends to decrease when the COP value is increased. Similarly to our result, the payback period ranges between four and five years when the COP value is 4 [44]. Owing to the high depreciation time, livestock farmers can implement an AHP-based livestock housing system to minimize their electricity costs, and it has the potential to work for a longer period.

## 5. Conclusions

In global intensive livestock farming systems, higher electricity consumption and inadequate air quality adversely influence the environmental sustainability and slow productivity. Therefore, the implementation of innovative strategies in order to maintain production parameters while reducing energy consumption and providing proper air quality is a current issue and is worthy of being collaboratively investigated. The present study aimed to investigate an AHP system to be utilized for intensive pig farming as an efficient, eco-friendly alternative to the widely used conventional electric heating systems. According to the results of this study, the inside temperature was maintained at a significantly higher level in the AHP house. A significant decrease in average electricity consumption by 40 kWh, overall cost, and $CO_2$ emissions by 19.32 kg was observed during the experimental period in the AHP house. Furthermore, the $NH_3$ and $H_2S$ emissions were also lower in the AHP-installed house than in the house with the conventional electric heating system. Although the initial installation cost was high, the investor could obtain long-term benefits with a uniform performance for a longer period (approximately 15 years) while utilizing less electricity and causing less greenhouse gas (GHG) emissions. Therefore, the AHP system is an innovative and sustainable energy source for cost-effective and eco-friendly heating of animal houses in the livestock sector.

**Author Contributions:** Conceptualization, C.J.Y. and K.W.P.; methodology, M.G.J. and H.S.M.; writing—Original draft preparation, D.R. and M.G.J.; software, D.R. and H.S.M.; data curation, S.R.L. and M.A.D.; formal analysis, C.J.Y. and K.W.P.; investigation, C.J.Y. and H.S.M.; writing—Review and editing, M.A.D., S.R.L., and D.R.; supervision, C.J.Y. and H.S.M., project administration, C.J.Y. and K.W.P.; funding acquisition, S.R.L. and M.G.J. All authors have read and agreed to the published version of the manuscript.

**Funding:** This study was funded by the Industrial Technology Innovation Business (20194210100020, Development and Demonstration of Renewable Energy Mixed-Use System for the Livestock Industry) and Ministry of Trade, Industry, and Energy, Korea.

**Conflicts of Interest:** The authors declare no conflict of interest.

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
