# Peer review of "Effect of a Sustainable Air Heat Pump System on Energy Efficiency, Housing Environment, and Productivity Traits in a Pig Farm"

_sustainability, doi:10.3390/su12229772_

Round 1
Reviewer 1 Report
By comparing the modified version with the first submitted manuscript, I found that there is a great change in the discussion section. Why do you make these changes? And the title of the manuscript was altered as well. It is recommended to use the track change function in the word processing software to clearly identify the modified area in the manuscript including those deleted descriptions. Furthermore, any alternation or omission of the content should have an explanation as a responsible research manner.
Reviewer 2 Report
This revised manuscript adds nothing valuable to the original paper. Only a couple of obvious mistakes have been corrected, but the core faults of the study still remain. I am very sorry, but I cannot accept this paper for publication in Sustainability.
Specifically:
- the entire study is built on a design error: first of all, the two rooms face two different directions (north and south). They accordingly receive very different solar radiation amounts, which are well known to be critical for heat transfer and pollutants movement in the room via air convection.
Secondly, the two rooms are kept at different temperatures (the AHP room had higher indoor air T), as remarked e.g. at line 254 “the inside temperature was maintained at a significantly higher level in the AHP house.”. This is accordingly not a fair 1 to 1 comparison. In other words, these two rooms are not comparable, especially regarding pollutants monitoring.
Qualitatively, it is evident that the AHP is way more efficient than any traditional electric heating, so we did not need a study to confirm this. Quantifying how much of a gain one can get, in terms of energy consumption, indoor air quality, pigs wellbeing etc., under the same indoor conditions should instead be the result of this work. Unfortunately, this is not accomplished here.
- in every table the SEM is still too large (often larger than 50%, sometimes even reaching the measured value) for any reasonable conclusions. If the statistical analysis is correct, this means that the measurements need to be performed again, the current results cannot be published in a scientific Journal.
Reviewer 3 Report
The submitted manuscript has been significantly improved. Although the authors have made major changes, I still consider some points unresolved.
- Please check all 6 occurrences of the term "renewable" in your article and remove it when AHP is associated. I suggest to use “sustainable” instead of “renewable”. The energy source for AHP is electricity produced mainly from fossil fuels. On the other hand, it has a relatively low carbon footprint.
- Introduction:
- Sustainability: See please my comment 1 and use sometimes the word “sustainable”.
- The novelty of the work: Thank you for the explanation. Please extend the introduction in the same sense. Emphasize in particular the fact that there are not many publications on this topic.
- Materials and methods:
- My question was, if is the electric heating system really “conventional” in South Korea. Thank you for your response. Please put it in the manuscript.
- My second comment was connected with the fact, that all the results in chapter 3.3 related to ventilation. They are not connected at all with the heat source (heat pump, gas boiler, solar heating ... – does not matter.). If you add the ventilation to a pig farm, you will get the results of chapter 3.3. Please make this clear in the text. At the moment, it looks like change of the heat source can improve the emissions in the interior, which is, of course, nonsense. Everything was caused by ventilation, which was not part of the original (convetional) system. Wherever you are talking about the levels of NH3, H2S, ultrafine dust (PM2.5) and formaldehyde in the manuscript, please state that the improvement is the result of ventilation (which is part of the new AHP system).
- Ok.
- Done Ok.
- Specify the price of electricity (l. 118). Please explain this (KEPCO) in the text, not just me. Add a date, on which the price is valid (September 2020?).
- Other minor comments:
- Check the equation on l. 113 – two dots above Q; I think the termp “capacity” is not correct. Use please “useful heat extracted from the heat pump (condensor)“.
- Table 6 – do not use the decimal places for costs and savings in USD, esp. when it is per annum. So instead 4,323.85 just 4,323 please. etc.
- Check the references on lines 368 – 377 (41 – 43?). Use square brackets on line 250.
Round 2
Reviewer 1 Report
The track-change notes on the right-hand side of the manuscript’s layout are all truncated perhaps due to the file format conversion process in the revised uploaded version of the manuscript in PDF format. Therefore, it’s still hard to differentiate what had been changed in the previous manuscript.
Reviewer 2 Report
I thank the authors for the revision, now I can finally understand the methodology and results. The previous manuscript versions omitted important information, and when this was already present, the low writing quality made it difficult to understand.
Comments 1-2: The added explanations, such as the discussion on the temperature setpoints at lines 180-185 and the new Figure 3, do clear out most of the inconsistencies I pointed out before. I accordingly believe that the main problem mostly concerned the exposition, not methodology nor results.
Comment 3:
- "While we appreciate the reviewer’s feedback, we respectfully disagree. We were carefully re-run our statistical analysis and we found two mistakes in NH3 SEM value and they were corrected (line 202), one mistake of formaldehyde SEM value and corrected it (line 211)."
This is not an answer to my criticism, as those corrections were already present in v1. All in all, many SEMs regarding the pollutants results are too large for significant results; however this is a preliminary study and we could make ouselves content with it. I strongly encourage focusing on higher precision in future studies.
----
Anyway, the present v2 of the manuscript contains remarkable improvements indeed, and for a preliminary study I deem the results significant enough for a publication.
- Please perform a full revision of the English language, some statements are really difficult to understand besides being grammatically wrong.
- Please increase the font size in each and every plot in Figure 3.
Round 3
Reviewer 1 Report
In this resubmitted version (v3), as compared to the first submitted version, I still cannot track what has been erased and what has been added. May I kindly ask that please use the track change function in the word processing software?
This manuscript is a resubmission of an earlier submission. The following is a list of the peer review reports and author responses from that submission.
Round 1
Reviewer 1 Report
There is one logical problem exhibited in this research, the indoor temperature of the pig houses in both the AHP and the conventional one is not kept identical. When discussing the benefits of AHP, the discussion should be made under the same indoor climate condition. Moreover, the AHP pig house and the conventional pig house are not in the same orientation, this could result in that the exterior walls will receive different solar radiation heat gains and have different amounts of air infiltration. The various amounts of infiltration will thereby influence the removal rate of the PM2.5 and other air quality-related index. Under this circumstance, the experimental results can hardly be comparable. The conclusion states that the inside temperature of AHP installed pig house was maintained at a significantly higher level than the conventional one, this also implies the incapability of the heating system, the reason may due to the fact of the insufficient capacity of the heating system of the traditional one. Therefore, the comparison is not fair.
Furthermore, on Line 113, the equation of COP seems incorrect, the denominator should be the power consumption and the numerator should be the heating capacity generated by the heat pump.
Reviewer 2 Report
The study covers an interesting topic towards sustainability, and it is rather clear in the structure and exposition. Unfortunately, despite the massive literature review (which is also misplaced in the Discussion section), in my opinion the scientific merit of the study is very low, and the investigation needs to be reassessed.
Maybe I have misinterpreted the data analysis results, otherwise it seems to me that either the measurements or the statistical analysis are incorrectly performed. A much more exhaustive assessment of the results is needed, please see my comments below.
I do not deem the present manuscript acceptable for publication in Sustainability, but once those problems have been fixed it could become an interesting and useful investigation.
Comments/questions:
- Insert numbers for all equations.
- 1 at line 113 is incorrect and is also counterintuitive, as the ratio should be the inverse of what is reported.
- Table 3: the SEM is too large for NH3, during weaning and growing periods, therefore I do not think those data carry any real significance.
- How was the conventional electric heating system implemented inside the room? Where did you place the inlet of the plastic pipe connected to air heating? A more detailed description of the two rooms and of the according heating systems must be provided.
- Lines 172-173: “There were no significant differences in the PM 2.5 dust concentration between the two houses. On the other hand, the dust concentration tended to decrease during all periods, and the average reduction was 6.5% compared to the control house.”
I might have missed something, but these two statements seem mutually contradictory to me. Could you please explain what you meant here? - Table 4: even worse, here the error is always comparable to the observations. It is too large, making the statistical analysis meaningless. I therefore disagree with the statement “the formaldehyde concentration was significantly lower (p<0.05) in the AHP house.”, as considering the error would make the AHP value even larger than the control value.
- Line 195: “We speculated that the AHP system could distribute the uniform heat pattern continuously inside the house than the conventional electric heating system.”
There are no reasons given for supporting this claim, nor dedicated measurements or computer simulations. The heating system needs to be addressed properly to this aim. - The Discussion section contains plenty of references to the literature about the COP result, and this is good, however the core results of Section 3 are almost completely overlooked.
As a matter of fact, all this discussion could be moved to the Introduction, as it constitutes a literature review rather than a critical analysis of the study results. - Only lines 208-210, 253-258, 263-264 contain (very superficially) some comments on the results. Everything else is too verbose and not pertinent to a Discussion section, but to the Introduction.
